# NSC-derived exosomes enhance therapeutic effects of NSC transplantation on cerebral ischemia in mice

**Ruolin Zhang[1,2], Weibing Mao[3], Lumeng Niu[3], Wendai Bao[1], Yiqi Wang[1,2], Ying Wang[1], Yasha Zhu[3], Zhihao Yang[1,2], Jincao Chen[4], Jiawen Dong[1], Meng Cai[3], Zilong Yuan[5], Haikun Song[1], Guangqiang Li[1], Min Zhang[1], Nanxiang Xiong[4]\*, Jun Wei[3]\*, Zhiqiang Dong[1,2]\***

[1]College of Biomedicine and Health, College of Life Science and Technology, Huazhong Agricultural University, Wuhan, China; [2]Center for Neurological Disease Research, Taihe Hospital, Hubei University of Medicine, Shiyan, China; [3]iRegene Therapeutics Co., Ltd, Wuhan, China; [4]Department of Neurosurgery, Zhongnan Hospital of Wuhan University, Wuhan, China; [5]Department of Radiology, Hubei Cancer Hospital, Tongji Medical College, Huazhong University of Science and Technology, Wuhan, China

**\*For correspondence:**
13971139959@163.com (NX);
weijun@iregene.com (JW);
dongz@mail.hzau.edu.cn (ZD)

**Abstract** Transplantation of neural stem cells (NSCs) has been proved to promote functional rehabilitation of brain lesions including ischemic stroke. However, the therapeutic effects of NSC transplantation are limited by the low survival and differentiation rates of NSCs due to the harsh environment in the brain after ischemic stroke. Here, we employed NSCs derived from human induced pluripotent stem cells together with exosomes extracted from NSCs to treat cerebral ischemia induced by middle cerebral artery occlusion/reperfusion in mice. The results showed that NSC-derived exosomes significantly reduced the inflammatory response, alleviated oxidative stress after NSC transplantation, and facilitated NSCs differentiation in vivo. The combination of NSCs with exosomes ameliorated the injury of brain tissue including cerebral infarction, neuronal death, and glial scarring, and promoted the recovery of motor function. To explore the underlying mechanisms, we analyzed the miRNA profiles of NSC-derived exosomes and the potential downstream genes. Our study provided the rationale for the clinical application of NSC-derived exosomes as a supportive adjuvant for NSC transplantation after stroke.

## Editor's evaluation

The authors have demonstrated that NSC-derived exosomes could act as a supportive adjuvant for NSC transplantation after stroke. NSC-derived exosomes significantly reduced the inflammatory response, alleviated oxidative stress after NSC transplantation, and facilitated NSC differentiation in vivo. Overall, the combination of NSCs with exosomes ameliorated the injury of brain tissue including cerebral infarction, neuronal death, and glial scarring, and promoted the motor function recovery.

## Introduction

Stroke is the second leading cause of death worldwide, which usually causes motor and cognitive impairments that require long-term rehabilitation (*GBD 2019 Stroke Collaborators, 2021*). The commonly used treatments of stroke in clinic include tissue plasminogen activator thrombolytic

therapy and thrombus clearance surgery, but both are limited by inability to repair damaged neural circuits, and only 10% of stroke patients meet the treatment standards (*Fugate and Rabinstein, 2015*; *Nagaraja et al., 2020*). Stem cell-based therapy is a progressing and promising method to treat ischemic stroke. Many stem cell types including neural stem cells (NSCs) (*Sakata et al., 2012*), embryonic stem cells (ESCs) (*Meamar et al., 2013*), mesenchymal stem cells (*Toyoshima et al., 2017*), bone marrow mononuclear cells (*Yang et al., 2012*), and iPSCs (induced pluripotent stem cells; *Duan et al., 2021*) have been tested in preclinical and clinical research, which showed encouraging therapeutic effects. Both endogenous and exogenous NSCs have remarkable capacity to maintain self-renewal while differentiating into various cell types including neurons and glial cells in nervous system (*Dong et al., 2012*). iPSCs can be an ideal resource to acquire NSCs, which voids both ethical problems and immune rejection, and has a potential to provide genetically identical 'patient-specific' cells for stroke patients (*Baker et al., 2019*). On the other hand, the low survival rate of transplanted NSCs, largely due to chronic inflammation and oxidative stress of the microenvironment after stroke (*Koutsaliaris et al., 2022*), and the poor differentiation of NSCs (*Zhang et al., 2019*) limited its application.

NSC-derived exosomes are enriched in specific miRNAs that mediate multiple functions in physiological and pathological conditions (*Luo et al., 2022*). NSC-derived exosomes, and have been proven useful for treating multiple neurological diseases due to their anti-inflammatory, neurogenic and neurotrophic effects as well as the interaction with the microenvironment of the brain tissue (*Vogel et al., 2018*). Previous studies suggested that application of NSC-derived exosomes could promote the differentiation of NSCs through miRNAs in vitro (*Yuan et al., 2021*). However, the effect of exosomes on grafted NSCs in vivo remains elusive. We propose that the combined treatment of exosomes and NSCs can effectively ameliorate harsh lesion conditions to help the NSCs survival and differentiation, achieving optimal treatment effects.

In this study, we established ischemic stroke in mice with middle cerebral artery occlusion/reperfusion (MCAO/R), and tested different treatment strategies using transplantation of iPSC-induced NSCs and NSC-derived exosomes. Our results indicated that NSC-derived exosomes could promote NSCs differentiation, reduce oxidative stress and inflammation, and alleviate the formation of glial scars after ischemia and reperfusion, and as a result, could enhance the therapeutic effects of NSC transplantation. We further explored the molecular mechanisms through profiling the miRNAs of the NSC-derived exosomes.

## Results

### NSC-derived exosomes facilitate post-stroke recovery after NSC transplantation in MCAO/R mice

We first characterized the NSCs derived from iPSCs by examining the expression of NSC marker genes including *SOX2* and *PAX6* by immunocytochemistry staining. The results showed that the cells used for subsequent transplantation expressed high level of NSC marker genes (*Figure 1—figure supplement 1A*), indicating that NSCs were efficiently induced from iPSCs. We isolated exosomes from the same NSCs and examined the expression of exosomal markers including TSG101, CD63, and CD9 (*Figure 1—figure supplement 1B*). Furthermore, the results of transmission electron microscopy showed that the particle size of exosomes mixture was less than 200 nm, and nanoparticle tracking analysis confirmed the typical distribution of particle diameter of exosomes (*Figure 1—figure supplement 1C, D*).

We next examined the effects of different treatment strategies on the brain lesion after cerebral ischemia and reperfusion in MCAO/R mice. To examine the presence and persistence of cerebral edema, 2,3,5-triphenyl tetrazolium chloride (TTC) staining was performed 1 and 7 days after MCAO/R (*Figure 1—figure supplement 1E*). Two doses of NSC transplantation, $2 \times 10^5$ and $5 \times 10^5$, were first tested. The results of survival analysis (*Figure 1—figure supplement 1F*) and rotarod test (*Figure 1—figure supplement 1G*) showed the dose-dependent effects of transplanted NSCs. Therefore, we determined to use the dose of $5 \times 10^5$ NSCs for the subsequent treatments to achieve a robust therapeutic effect. Mice were randomly divided into five groups (Sham, PBS, Exo, NSC, and NSC + Exo). Except Sham group, mice in all the other four groups received standard MCAO/R surgery. Lateral ventricle injections of 5 µl PBS (PBS group), 10 µg exosomes in 5 µl PBS (Exo group), $5 \times 10^5$ NSCs in 5 µl PBS (NSC group), and $5 \times 10^5$ NSCs + 10 µg exosomes in 5 µl PBS (NSC + Exo

group) were performed at 7 days post-MCAO/R (*Figure 1A*). The levels of reactive oxygen species (ROS) and inflammation were measured in focal brain tissues at 3 days post treatment; behavioral assessments were performed at 0-8 weeks post treatment; histological examinations were analyzed at 8 weeks post treatment (*Figure 1A*). To ensure the successful establishment of cerebral ischemia, the cerebral blood flow was examined before, during, and after MCAO/R (*Figure 1B*). Neurological functions were evaluated by balance beam, ladder lung, rotarod test, and modified neurological severity score (mNSS) up to 8 weeks after treatment (*Figure 1C* and *Figure 1—figure supplement 2A*). The results suggested that transplantation of NSCs combined with exosomes began to take effect starting at 4 weeks after treatment and significantly worked better than that solely with NSCs at 8 weeks post treatment (*Figure 1C*). The infarct area in the ipsilateral hemisphere was determined by MRI (*Figure 1D*) at 8 weeks post treatment. Compared to the severe damages of brain tissues in PBS group, mice treated by NSCs combine with exosomes showed significantly reduced infarct areas (*Figure 1E*). Meanwhile, the combination of NSCs and exosomes showed better protective effects on the brain tissue than either alone (*Figure 1F*), which was further confirmed by the results of brain weight analysis (*Figure 1G*). Therefore, our results indicated that NSC-derived exosomes could significantly enhance the therapeutic effects of NSCs on motor dysfunction and brain infarction in MCAO/R mice. Furthermore, the NSC-mediated therapeutic effects were greatly accelerated by addition of exosomes.

## NSC-derived exosomes enhance the therapeutic effects of NSCs on neuronal damage

We next examined the recovery of ischemia-induced neuronal damage of cerebral cortex in different treatment groups. The results of immunostaining and qRT-PCR of RBFOX3/NeuN revealed that, compared with the mice treated solely with NSCs, the combination NSCs and exosomes significantly reduced the tissue loss from 14.32 ± 3.52% to 7.57 ± 2.59% (*Figure 2A, B*; *Figure 2—figure supplement 1A*). Consistently, Nissl staining showed that MCAO/R mice had damaged pyramidal and granular cells with fuzzy cell contours (*Figure 2C* and *Figure 2—figure supplement 1B*). The addition of exosomes could further reduce the neuronal loss in the ipsilesional hemisphere on top of the effects of NSC transplantation. To further explore the effects of exosomes on neuronal survival, HT22 cells were subjected to oxygen and glucose deprivation (OGD)/reoxygenation (OGD/R). Immunostaining on neuronal marker MAP2 and apoptotic marker cleaved Caspase-3 (c-Caspase-3) showed that OGD/R caused robust neuronal apoptosis was alleviated by exosomes (*Figure 2—figure supplement 1C*). Western blot results showed that the expression of Caspase-3 and c-Caspase-3 was significantly increased after OGD/R, whereas exosomes downregulated the expression of c-Caspase-3 (*Figure 2—figure supplement 1D*). We further examined the effects of exosome treatment on the mRNA expression level of pro-apoptotic gene *Bax* by qRT-PCR, which confirmed that exosome significantly reduced the expression of *Bax* (*Figure 2—figure supplement 1E*).

We performed Golgi staining to examine the recovery of neuronal complexity. The results suggested that the reduction of infarct area after combined treatment was also accompanied with improved dendritic density and length (*Figure 2D, E*), alleviated spines loss (*Figure 2F*), and increased complexity of neuronal projections (*Figure 2G*) in the cerebral cortex. Interestingly, although exosome treatment did not show robust therapeutic effects on behavior impairment and infarct area, the number of dendritic spines was significantly increased by exosome treatment in Exo group compared to that of PBS group (*Figure 2F*), suggesting that exosomes might play an important role in the recovery of neuronal complexity. SYN1 is a member of the synapsin family, localizes on the presynaptic membrane, and plays a crucial role in the regulation of axonogenesis and synaptogenesis. We examined the expression of SYN1 to by western blot to evaluate the recovery of neural connection, which showed that SYN1 expression was increased in both NSC and NSC + Exo groups at 8 weeks after treatment (*Figure 2H*).

## Exosomes promote the survival and differentiation of transplanted NSCs

We first examined the overall apoptosis in the cerebral cortex of mice after MCAO/R by TUNEL staining. The results revealed that the excessive cell apoptosis caused by MCAO/R was reduced by exosome transplantation (*Figure 3—figure supplement 1B*). As NSCs and exosomes were

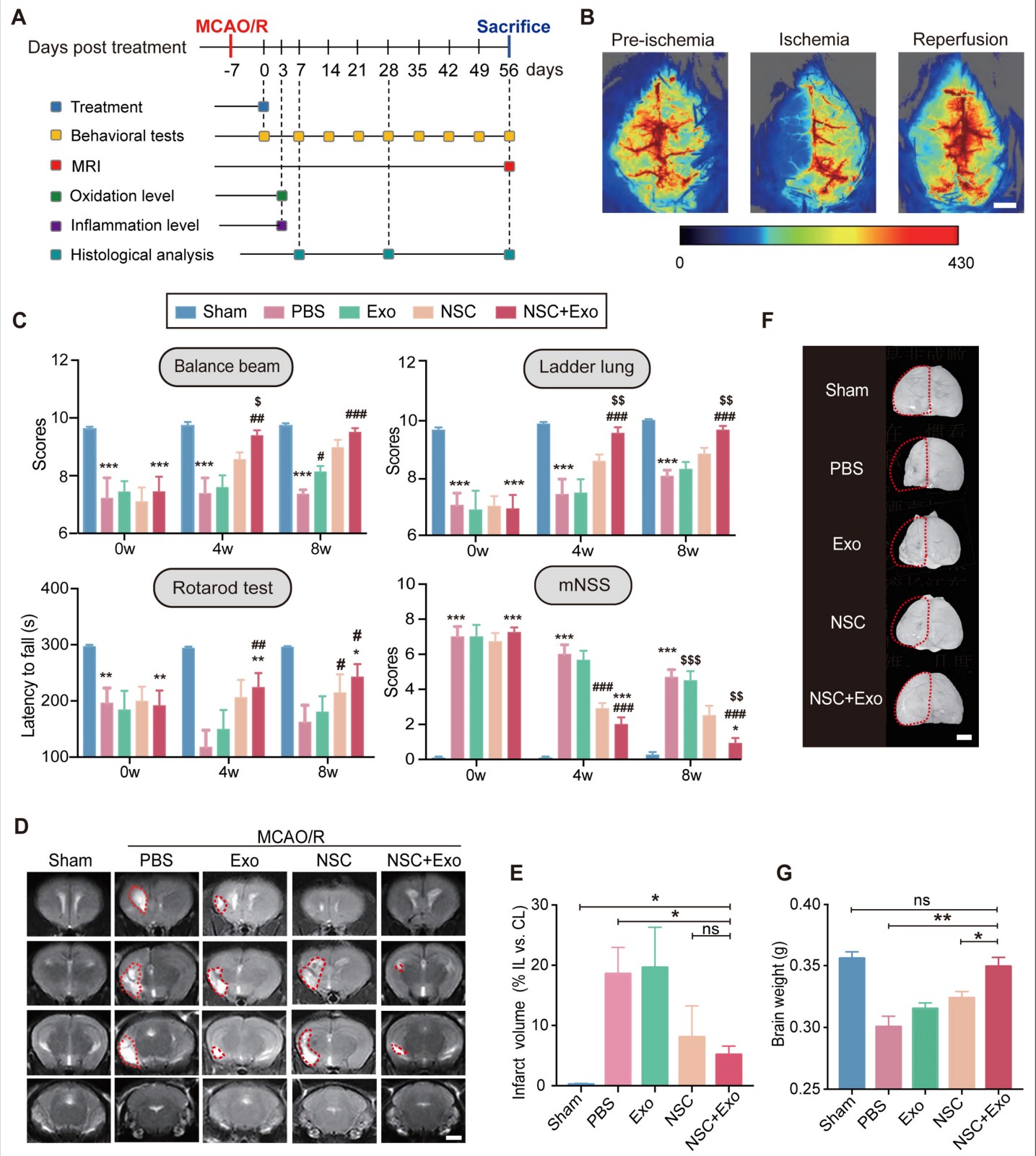

**Figure 1.** Neural stem cell (NSC)-derived exosomes enhanced the therapeutic effects of NSCs on motor impairment and brain infarction after stroke. (**A**) Summary of the experimental timeframes. (**B**) Images of cerebral blood flow before, during, and 24 hr of the middle cerebral artery occlusion/ reperfusion (MCAO/R) procedure. Scale bar: 2 mm. (**C**) Behavioral test results (the balance beam, ladder rung, rotarod tests) and modified neurological severity score (mNSS) at 0, 4, and 8 weeks after treatment, $n$ = 10 mice per group. *$p < 0.05$, **$p < 0.01$, ***$p < 0.001$, versus Sham group. #$p < 0.05$, ##$p$

*Figure 1 continued on next page*

*Figure 1 continued*

< 0.01, ###p < 0.001, versus PBS group. $p < 0.05, $$p < 0.01, $$$p < 0.001, versus NSC group. (**D**) MRI images show brain cerebral infarct at 8 weeks after treatment. The infarct area is marked by dotted lines. Scale bar: 2 mm. (**E**) Quantification of (**D**), n = 3 per group. (**F**) Representative images show brain atrophy at 8 weeks after treatment. The ischemic hemispheres are marked by dotted lines. Scale bar: 2 mm. (**G**) Quantification of the brain weights at 8 weeks after treatment, n = 6 per group. *p < 0.05, **p < 0.01, ns indicates non-significant difference.

The online version of this article includes the following source data and figure supplement(s) for figure 1:

**Source data 1.** Neural stem cell (NSC)-derived exosomes enhanced the therapeutic effects of NSCs on motor impairment and brain infarction after stroke.

**Figure supplement 1.** Characterization of neural stem cells (NSCs) and NSC-derived exosomes, and effects of different doses of transplanted NSCs.

**Figure supplement 1—source data 1.** Characterization of neural stem cells (NSCs) and NSC-derived exosomes, and effects of different doses of transplanted NSCs.

**Figure supplement 1—source data 2.** Characterization of neural stem cells (NSCs) and NSC-derived exosomes, and effects of different doses of transplanted NSCs.

**Figure supplement 2.** Effects of different treatment strategies of neural stem cells (NSCs) and exosomes on ischemic stroke in middle cerebral artery occlusion/reperfusion (MCAO/R) mice.

**Figure supplement 2—source data 1.** Effects of different treatment strategies of neural stem cells (NSCs) and exosomes on ischemic stroke in middle cerebral artery occlusion/reperfusion (MCAO/R) mice.

transplanted simultaneously into the lateral ventricle (***Figure 3—figure supplement 1A***), we next inspected whether exosomes could inhibit the apoptosis of transplanted NSCs. The mice were sacrificed and the brain isolated for analysis 1 week after transplantation. To track the transplanted NSCs, we employed the human-specific STEM121 antibody, which enables the quantification of engraftment, survival, migration, and differentiation of transplanted human stem cells in xenograft models. Serial sections covering the transplantation zone were stained for c-Caspase-3 and STEM121. As shown in ***Figure 3A***, co-transplantation of exosomes reduced the number of c-Caspase-3$^+$/STEM121$^+$ cells, suggesting that exosomes promote the survival of transplanted NSCs. Interestingly, the anti-STEM121 staining revealed a larger distribution area of STEM121$^+$ cells in NSC + Exo group compared with NSC group suggesting that exosomes might help the migration of transplanted cells (***Figure 3—figure supplement 1C***).

To investigate the regulatory effects of exosomes on the differentiation of transplanted NSCs, cerebral sections of mice at 4 and 8 weeks after transplantation were immunostained with specific antibodies for NSC and neuronal markers to assess the extent of NSC differentiation. Nestin is a NSC marker and its expression is downregulated once NSCs start to differentiate (***Park et al., 2010***). Tuj1 is a neuronal marker from the early stage of neural differentiation (***Nogueras-Ortiz et al., 2020***). RBFOX3/NeuN is detected exclusively in post-mitotic mature neurons (***Gusel'nikova and Korzhevskiy, 2015***). Postsynaptic density protein-95 (PSD95) is a scaffolding protein involved in the assembly and function of the postsynaptic density complex (***Mardones et al., 2019***). The results showed that, at 4 weeks after transplantation, NeuN and PSD95 staining were rare in STEM121$^+$ cells in NSC group while NeuN$^+$/STEM121$^+$ and PSD95$^+$/STEM121$^+$ cells were significantly increased in NSC + Exo group (***Figure 3B*** and ***Figure 3—figure supplement 1D***). The results suggest that NSCs rarely differentiate at 4 weeks after transplantation, which is promoted by exosome treatment. We further analyzed the NSC differentiation at 8 weeks. Compared with the NSC group, the number of tdTomato-positive NSCs was significantly increased in the NSC + Exo group (***Figure 3C, E***). Among the tdTomato-positive cells, Nestin$^+$/tdTomato$^+$ cells were less in NSC + Exo group than the other groups (***Figure 3C, F***), while the number of Tuj1$^+$/tdTomato$^+$ cells was significantly higher in NSC + Exo group, which implied that exosomes could promote the differentiation of NSCs into neurons (***Figure 3D, G***). Therefore, our data indicated that co-transplantation of exosomes could effectively facilitate the differentiation of transplanted NSCs in MCAO/R mice.

## Exosomes promote the microenvironment remodeling

Oxidative stress and global brain inflammation are closely involved in the progressing pathology after stroke (***Hum et al., 2007***, ***Shi et al., 2019***), which challenges the survival and colonization of transplanted NSCs (***Li et al., 2017***). We employed OGD/R on cultured NSCs to simulate the main pathogenesis of stroke, ischemia–reperfusion (***Zhang et al., 2017***; ***Yu et al., 2018***). The results showed that

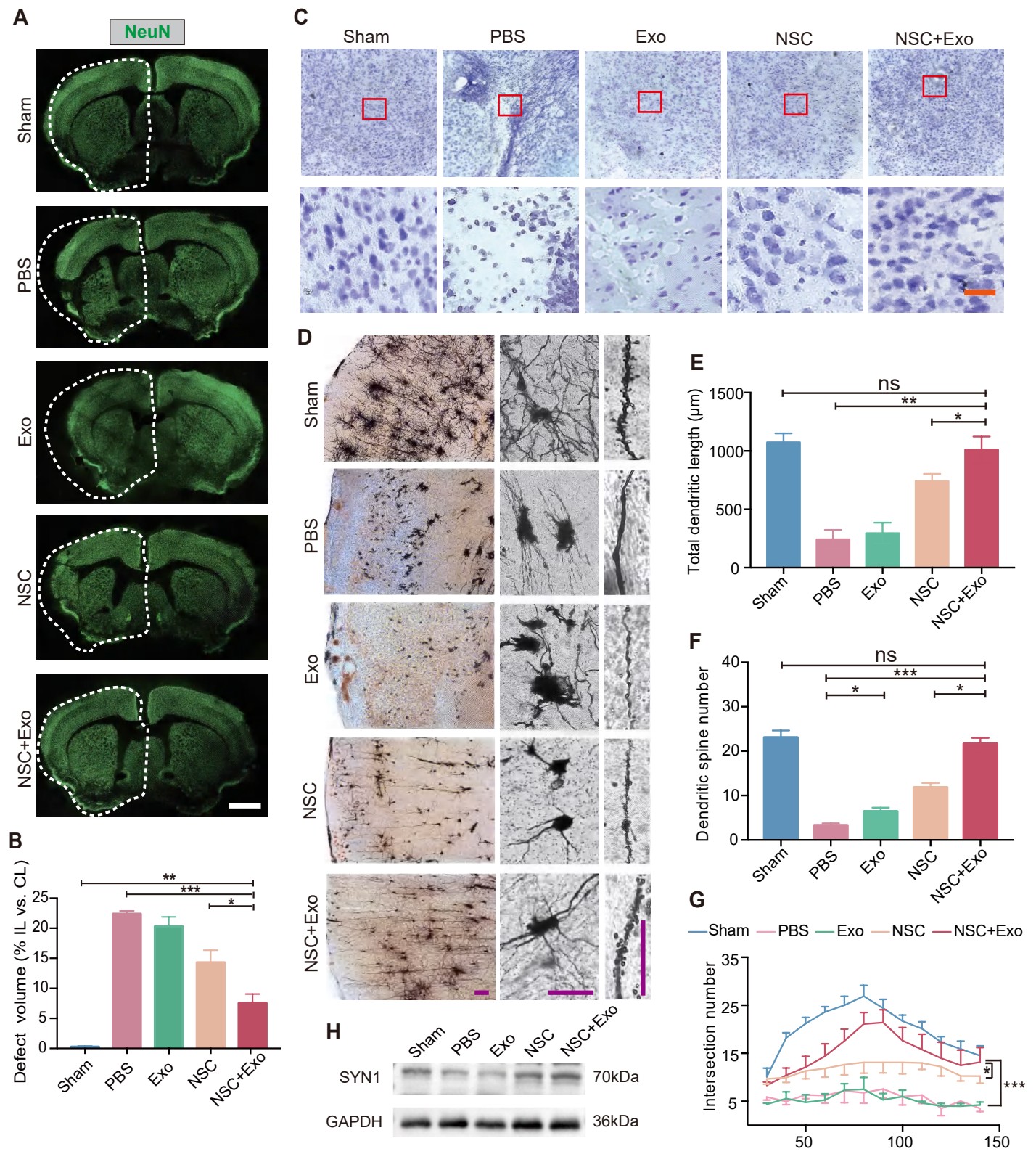

**Figure 2.** Effects of combined treatment with neural stem cells (NSCs) and exosomes on neuronal damage in middle cerebral artery occlusion/reperfusion (MCAO/R) mice. (**A**) Immunofluorescent staining of NeuN in different groups at 8 weeks after treatment. The ipsilateral hemispheres are marked by the dotted lines. Scale bar: 2 mm. (**B**) Quantification of defect volume of NeuN staining, $n$ = 4 mice per group. (**C**) Nissl staining of infarct area in the brain at 8 weeks after treatment. Scale bar: 25 µm. (**D**) Representative images of Golgi-Cox staining in the infarct area at 8 weeks after

*Figure 2 continued on next page*

*Figure 2 continued*

treatment. Quantitative analysis of total dendritic length (**E**), dendritic spine number (**F**), and neuronal complexity (**G**). Scale bar: 100 μm. Fifteen neurons from three mice were analyzed for each group. (**H**) Western blot results show the expression of SYN1 in the ipsilateral cerebral cortexes from different groups at 8 weeks after treatment, n = 3 per group. *p < 0.05, **p < 0.01, ***p < 0.001. ns indicates non-significant difference.

The online version of this article includes the following source data and figure supplement(s) for figure 2:

**Source data 1.** Effects of combined treatment with neural stem cells (NSCs) and exosomes on neuronal damage in middle cerebral artery occlusion/reperfusion (MCAO/R) mice.

**Figure supplement 1.** Neural stem cells (NSCs) and exosomes combination treatment reduced the neuronal loss in the ipsilesional hemisphere.

**Figure supplement 1—source data 1.** Neural stem cell (NSC)-derived exosomes enhance the therapeutic effects of NSCs on neuronal damage.

OGD/R treatment could induce high level of oxidative stress in NSCs, whereas exosomes could reduce the production of ROS after OGD/R (*Figure 4A, B*). We further examined the expression of oxidative stress-related genes. The mRNA expression level of *CHOP* (endoplasmic reticulum stress marker) was reduced by exosome treatment after OGD/R (*Figure 4C*). Meanwhile, exosome treatment increased the expression of antioxidant genes *NRF2*, *NQO1*, and *SOD2* (*Figure 4C*). Besides the in vitro OGD/R experiments, the level of oxidative stress in vivo was also determined at 3 days after MCAO/R, which disclosed that the MDA content was significantly decreased in exosome-treated mice (*Figure 4D*). Therefore, our data suggested that NSC-derived exosomes could ameliorate oxidative stress, which could potentially facilitate the survival, colonization and differentiation of transplanted NSCs.

Heterologous stem cells transplantation could induce robust inflammatory response. It has been reported that the proliferation of immune cells reaches the peak during the acute phase post-transplantation (*Graf and Stern, 2012*; *Boncoraglio et al., 2019*). Interestingly, our results showed that exosomes could reduce the expression of inflammatory cytokines including *Tnfa* and *Il1b*, while increase the expression of anti-inflammatory cytokine *Il10* in brain tissues (*Figure 4E*) suggesting that exosomes could alleviate the elevated immune response after NSC transplantation.

Inflammatory cytokines can induce the activation of A1 reactive astrocytes, after brain tissue damages caused by conditions such as cerebral ischemia and reperfusion. A1 reactive astrocytes produce complement components and release toxic factors which promote neuronal death (*Clarke et al., 2018*). A1 reactive astrocytes could form glial scars to reestablish the physical and chemical integrity of the brain tissue by generating a barrier across the injured area, but inhibit the neuronal recovery as well (*Michinaga and Koyama, 2021*). In order to explore the effects of NSC-derived exosomes on the generation of A1 reactive astrocytes, exosomes were applied to cultured astrocytes following OGD/R, and the expression of A1 astrocyte markers was examined by qRT-PCR. OGD/R significantly increased the expression of *C3*, *Gbp2*, and *Lcn2* at 24 and 48 hr post OGD/R, which was alleviated by exosomes (*Figure 4—figure supplement 1B, C*). Our results indicated that astrocytes were prone to form glial scars during the chronic phase after stroke in vivo (*Figure 4F–H* and *Figure 4—figure supplement 1A*). We subsequently investigated the effects of different treatments on the formation of glial scars in MCAO/R mice. The results suggested that the combined treatment of NSCs and exosomes significantly decreased the glia scars in the subacute phase (*Figure 4—figure supplement 1D*) and the chronic phase (*Figure 4F, H*).

## miRNA profiling and functional enrichment analysis of NSC-derived exosomes

To explore the underlying molecular mechanisms of exosomes regulating the transplanted NSCs and the brain microenvironment, we proposed that the exosomes might regulate target genes through the release of miRNAs, components of the key functional molecules carried by exosomes. Therefore, we profiled the miRNA expression of NSC-derived exosomes using miRNA microarray. A total of 850 known miRNAs were detected, and the top 10 miRNAs with the highest read counts were displayed and verified by qRT-PCR (*Figure 5A* and *Figure 5—figure supplement 1A*). Targetscan, miRcode and miRDB databases were used to predict the downstream targets of the top 10 abundant miRNAs, and 17 potential target genes were selected, which have been proven to play important roles in neural modulation. The interactive network of the exosomal miRNAs and the selected target genes were analyzed and visualized using Cytoscape (*Figure 5B*). Target genes were predicted to be regulated by multiple miRNAs, among which hsa-miR-30a-5p and hsa-miR-7-5p were involved in multiple

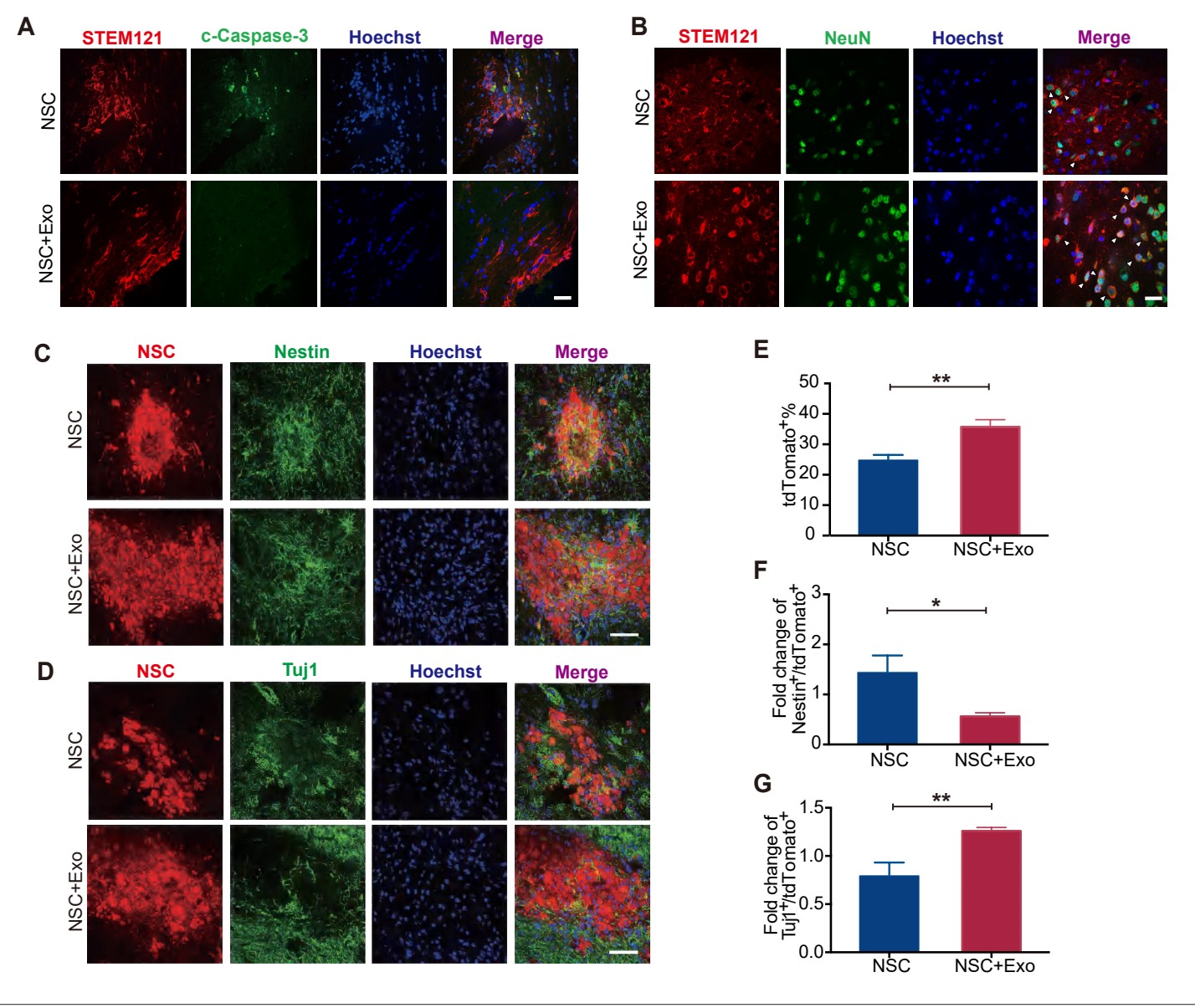

**Figure 3.** Effects of neural stem cell (NSC)-derived exosomes on the differentiation of transplanted NSCs. (**A**) Immunofluorescent staining of STEM121 and cleaved Caspase-3 (c-Caspase-3) at 7 days after transplantation. Scale bar: 20 μm. (**B**) Immunofluorescent staining of STEM121 and NeuN staining at 4 weeks after transplantation. Scale bar: 10 μm. White arrowhead: STEM121$^+$/NeuN$^+$ cells. (**C**) Representative images of Nestin staining at 8 weeks after transplantation. (**D**) Tuj1 staining at 8 weeks after transplantation. Quantification of tdTomato$^+$ cells (**E**), Nestin$^+$/tdTomato$^+$ cells (**F**), and Tuj1$^+$/ tdTomato$^+$ cells (**G**). Scale bar: 50 μm. *p < 0.05, **p < 0.01.

The online version of this article includes the following source data and figure supplement(s) for figure 3:

**Source data 1.** Neural stem cell (NSC)-derived exosomes promoted the differentiation of transplanted NSCs.

**Figure supplement 1.** Exosome transplantation reduced the excessive apoptosis cells after middle cerebral artery occlusion/reperfusion (MCAO/R).

regulation. We next examined the effects of exosome treatment on the expression of candidate target genes *STAT3*, *CHUK* (IKKα), and *PTPN1* (*Park et al., 2012*; *Wang et al., 2018*; *Culley et al., 2019*) in cultured NSCs after OGD/R by RT-qPCR and in MCAO/R mice brain tissues by western blot. The results confirmed that exosomes reduced the expression of downstream genes in NSCs (*Figure 5— figure supplement 1B*) as well as in brain tissues (*Figure 5—figure supplement 1C, D*), suggesting that exosomes might modulate the recipient cells and brain tissues through carrying miRNAs that regulate the expression of target genes.

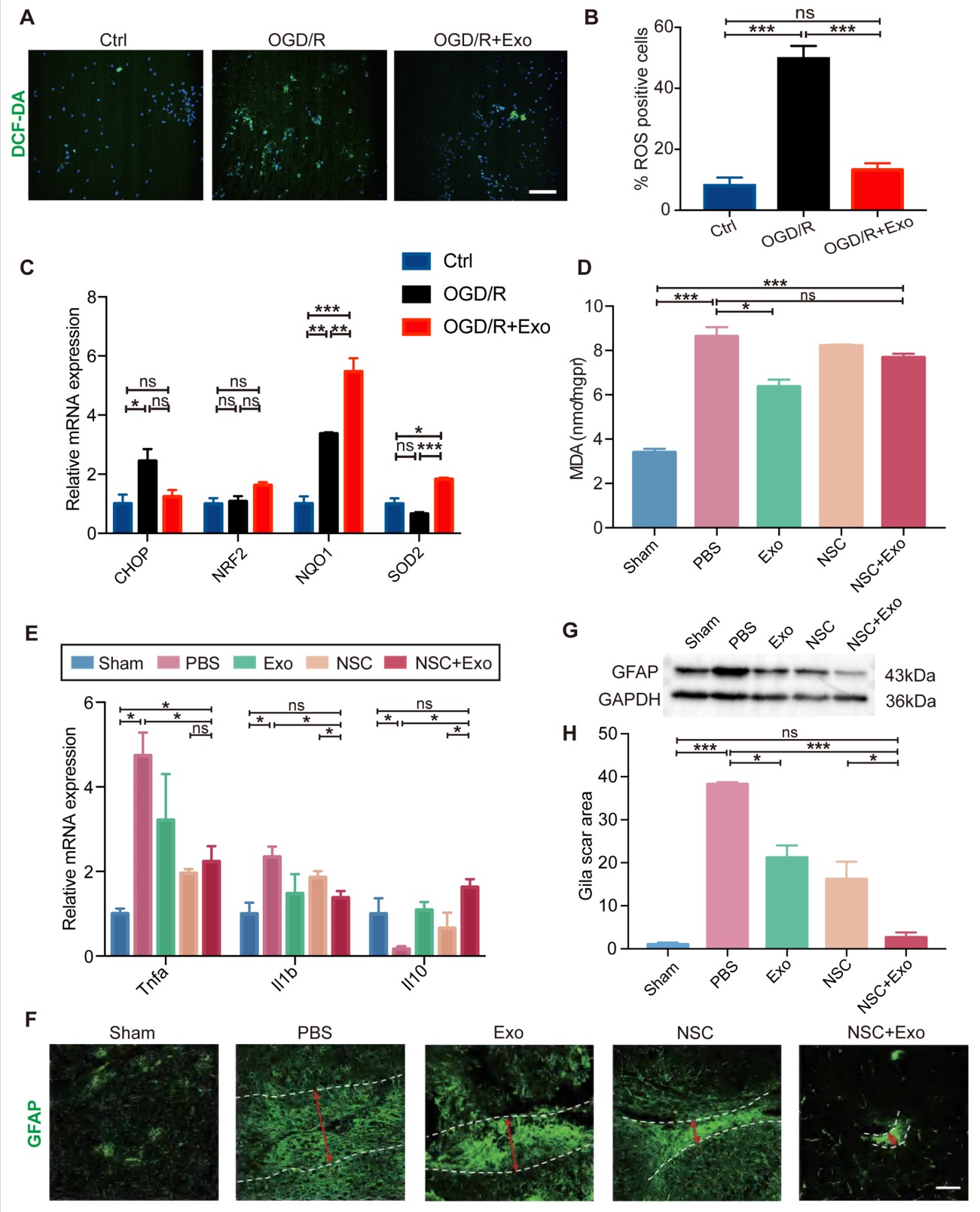

**Figure 4.** Effects of exosomes on the microenvironment remodeling. (**A**) Reactive oxygen species (ROS) generation was evaluated by DCF-DA fluorescent probe labeling (green) in oxygen and glucose deprivation/reoxygenation (OGD/R)-treated neural stem cells (NSCs). Nuclei were counterstained with Hoechst (blue). Scale bar: 50 μm. (**B**) Percentage of ROS-positive cells. (**C**) The relative mRNA expression of oxidative stress-related genes *CHOP*, *NRF2*, *NQO1*, and *SOD2* were measured by qRT-PCR. (**D**) The MDA level at 3 days after treatment, *n* = 3 mice/group. (**E**) The mRNA

*Figure 4 continued on next page*

*Figure 4 continued*

expression of *Tnfa*, *Il1b*, and *Il10* of the ipsilesional brain was measured by qRT-PCR at 3 days after treatment, *n* = 3 mice/group. (**F**) Representative images and quantification (**H**) of scar-forming astrocytes detected by GFAP staining. Scale bar: 50 μm. (**G**) Western blot results show the expression of GFAP in different groups. *p < 0.05, **p < 0.01, ***p < 0.001. ns indicates non-significant difference.

The online version of this article includes the following source data and figure supplement(s) for figure 4:

**Source data 1.** Effects of exosomes on the microenvironment remodeling.

**Figure supplement 1.** Combined treatment of neural stem cells (NSCs) and exosomes reduced the formation of glial scars in middle cerebral artery occlusion/reperfusion (MCAO/R) mice.

**Figure supplement 1—source data 1.** Combined treatment of neural stem cells (NSCs) and exosomes reduced the formation of glial scars in middle cerebral artery occlusion/reperfusion (MCAO/R) mice.

To further depict the regulatory effects of exosomal miRNAs on NSCs and the microenvironment, we performed gene ontology (GO) enrichment analysis and KEGG pathway analysis on all the potential target genes. GO enrichment analysis, in terms of biological process (*Figure 5C*), molecular function (*Figure 5D*), and cellular component (*Figure 5E*), disclosed that the potential target genes were enriched in functions that were correlated with cellular and microenvironmental homeostasis of the central nervous system such as regulation of neuron death and neurogenesis, stress-activated protein kinase signaling cascade, cytokine receptor binding, and neuron spine. KEGG pathway analysis suggested that the target genes were mainly involved in inflammation and apoptosis-related signaling pathways (*Figure 5F*). Therefore, the predicted target genes of exosomal miRNAs were concentrated in the functions and pathways that could regulate the cellular behavior of transplanted NSCs as well as the microenvironment remodeling.

Taken together, our findings suggested that NSC-derived exosomes might regulate the transplanted NSCs and the surrounding microenvironment through carrying the miRNAs which could further modulate the downstream genes and pathways in both the NSCs and the surrounding cells (*Figure 6*).

## Discussion

Stem cell-based therapy is an emerging and promising method to treat stroke, due to its effectiveness in cell replacement, neuroprotection, angiogenesis, and modulation of inflammation and immune response (*Hao et al., 2014*), but poor survival and differentiation of grafted cells have limited its efficacy and application (*Jiang et al., 2019*). In the present study, we co-transplanted NSC-derived exosomes with NSCs in MCAO/R-induced cerebral ischemia in mice. Consistent with previous reports (*Wei et al., 2017*; *Zhang et al., 2019*), our results confirmed that NSCs could effectively promote the recovery of motor function post-stroke in mice. Importantly, we demonstrated that exosomes could promote the repairment of the damaged brain tissue as well as the functional recovery, enhance the differentiation of the grafted NSCs in the infarct area, reduce the oxidative stress and inflammation, and alleviate the formation of glial scars in MCAO/R mice. As a proof-of-concept study on the co-delivery of NSCs together with exosomes in the classical animal model of ischemic stroke, our study provided solid rationale supporting the application of exosomes during stem cell-based therapy. On the other hand, whether exosomes from different sources have similar effects on transplanted NSCs and how exosomes regulate other types of stem cells in vivo deem further exploration.

The pathological process of ischemia–reperfusion includes the generation of ROS, brain edema, and the increased levels of inflammation, which leads to the tough microenvironment for the transplanted stem cells to survive and differentiate (*Zhang et al., 2019*). In addition, transplanted allogeneic stem cells also exacerbate oxidative stress levels. *Yahata et al., 2011* found that transplantation of human hematopoietic stem cells triggers replication stress and induces increasing ROS levels in mice. Bone marrow mesenchymal stem cells transplantation has also been reported to increase oxidative stress levels in mouse muscle cells (*Liu et al., 2019*). Besides the oxidative stress, the chemokines released by macrophages and endothelial cells after stroke, such as chemokine group CXC ligand 1 (CXCL1), recruit peripheral immune cells to flood into the damaged brain, which causes immune-inflammatory damage (*Ormstad et al., 2011*). Meanwhile, transplantation of exogenous stem cells could aggravate the inflammatory response around infract area due to the immune rejection (*Hao et al., 2014*). Xia et al. reported that ESC-derived exosomes decrease the inflammatory response,

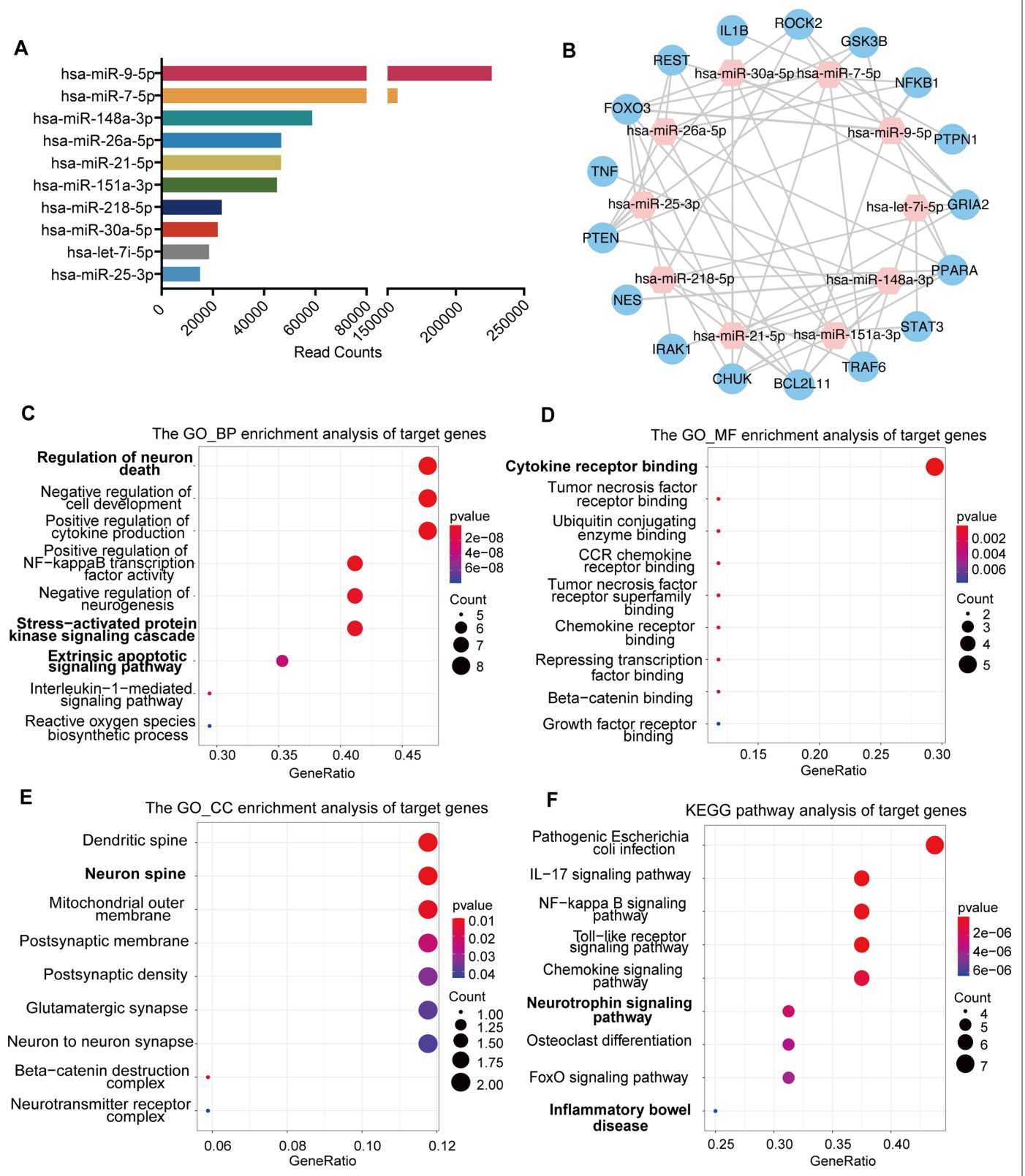

**Figure 5.** miRNA profiling of neural stem cell (NSC)-derived exosomes and enrichment analysis of predicted target genes. (**A**) The read counts of the top 10 abundant miRNAs in NSC-derived exosomes. (**B**) miRNA–mRNA regulatory networks. miRNA and mRNA are represented by the pink and blue circles, respectively. Gene ontology analysis of predicted target genes in terms of biological process (**C**), molecular function (**D**), and cellular component (**E**). (**F**) The KEGG pathway analysis of predicted target genes.

*Figure 5 continued on next page*

*Figure 5 continued*

The online version of this article includes the following source data, source code, and figure supplement(s) for figure 5:

**Source code 1.** Analysis of miRNA profiling of neural stem cell (NSC)-derived exosomes.

**Source code 2.** Gene ontology (GO) analysis of predicted target genes.

**Figure supplement 1.** Verification of top 10 abundant miRNAs and the expression of the selected target genes after oxygen and glucose deprivation/reoxygenation (OGD/R).

**Figure supplement 1—source data 1.** Verification of top 10 abundant miRNAs and the expression of the selected target genes after oxygen and glucose deprivation/reoxygenation (OGD/R).

alleviate neuronal death, and improve long-term recovery after MCAO/R through increasing regulatory T cells (*Xia et al., 2021*). Due to the properties of regulating signaling pathways in target cells, and remodeling the microenvironment (*Vogel et al., 2018*), NSC-derived exosomes have been demonstrated to improve a variety of neurological diseases, such as Alzheimer's disease (*Liu et al., 2020*), spinal cord injury (*Ma et al., 2019*), and ischemic stroke (*Sun et al., 2019*). Recent evidence demonstrated that exosomes promote the maturation of both neuron and glial cells in vitro (*Yuan et al., 2021*). Furthermore, excessive initiation of apoptosis has also been implicated in stroke (*Hwang et al., 2013*). Here, we showed that NSC-derived exosomes could reduce the oxidative stress and the inflammatory response, and promote the differentiation of transplanted NSCs and reduce excessive apoptosis in the brain of MCAO/R mice. Therefore, our results indicated that exosomes could promote the therapeutic effects of transplanted NSCs at multiple levels.

Previous studies have shown that stem cell-derived exosomes had neural protective effects and could promote recovery after ischemic stroke (*Webb et al., 2018*; *Sun et al., 2019*; *Xia et al., 2021*). However, we did not observe significant therapeutic effects with solely exosome treatment, which could be due to the dose of exosomes, the treatment timing and frequency. Considering the fact that cell transplantation requires a relatively stable microenvironment (*Nih et al., 2017*; *Lee et al., 2018*), we transplanted NSCs and exosomes at 7 days after stroke without subsequent delivery of exosomes in this study. Although the delivery of exosomes alone used in this study did not show significant neural protective effects, it indeed ameliorated oxidative and inflammatory lesion conditions, promoted neuronal repairment, and potentiated the therapeutic power of transplanted NSCs, suggesting that the application of exosomes could be an effective adjuvant for NSC-based therapy. Besides, as exosomes are ideal carriers for drug delivery (*Chen et al., 2021*), modifications of exosomes by adding drugs or other functional molecules could potentially further enhance the beneficial effects of exosome treatment.

As miRNAs were reported to be one of the major exosomal components, we profiled the miRNAs from NSC-derived exosomes to explore the molecular basis for the effects of exosomes as we observed in this study. Bioinformatic enrichment analysis in this study suggested that the predicted target genes

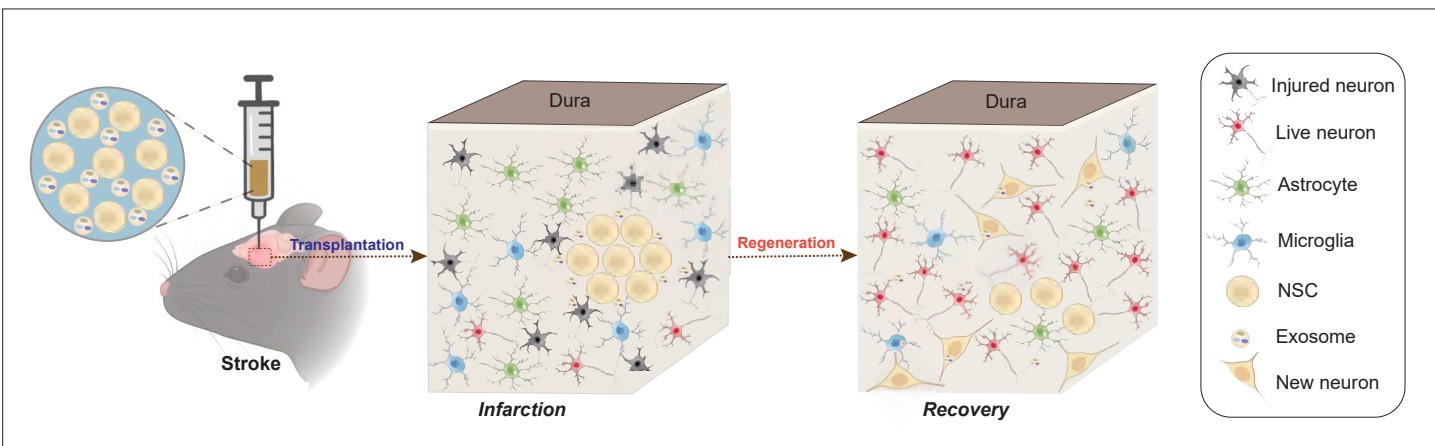

**Figure 6.** Schematic illustration of the mechanisms for combination treatment of neural stem cells (NSCs) and exosomes in neuroprotection against ischemic stroke. NSCs and exosomes achieve therapeutic goals by repairing damaged neurons, alleviating the inflammatory environment and reducing glial cell activation.

of exosomal miRNAs were concentrated in the functions and pathways that could regulate the NSCs' behavior and the surrounding microenvironment. Interestingly, inflammation and oxidative stress-related genes and signaling pathways were also highly enriched in the target genes, consistent with the antioxidant role of exosomes as disclosed by our study. We, therefore, provided clues and a useful resource of exosomal miRNAs and predicted target genes for understanding the mechanisms underlying the function of exosomes in the NSC-based therapy for ischemic stroke. The roles and working model of the exosomal miRNAs as well as predicted target genes demand further exploration.

# Materials and methods

## Key resources table

| Reagent type (species) or resource | Designation | Source or reference | Identifiers | Additional information |
|---|---|---|---|---|
| Cell line (*Homo sapiens*) | iPSCs | *Cai et al., 2021* | N/A | iPSCs |
| Commercial assay or kit | BCA assay | Biosharp | BL521A | |
| Chemical compound, drug | BSA | Sigma | N/A | |
| Chemical compound, drug | TTC | Solarbio | T8170 | |
| Antibody | Anti-Sox2 (Rabbit Polyclonal) | GeneTex | Cat. No. 43019 | 1:500 |
| Antibody | Anti-E-Cadherin (Mouse Monoclonal) | Proteintech | Cat. No. 60335 | 1:200 |
| Antibody | Anti-Pax6 (Rabbit Polyclonal) | Proteintech | Cat. No. 12323 | 1:200 |
| Antibody | Anti-Ki67 (Mouse Monoclonal) | Cell Signaling Technology | Cat. No. 9449 | 1:1000 |
| Antibody | Anti-NeuN (Rabbit Polyclonal) | Abcam | Cat. No. ab177487 | 1:300 |
| Antibody | Anti-MAP2 (Mouse Monoclonal) | Proteintech | Cat. No. 67015 | 1:500 |
| Antibody | STEM121 (Mouse Monoclonal) | Takara | Cat. No. Y40410 | 1:500 |
| Chemical compound, drug | Goat anti-Mouse IgG 568 | Invitrogen | A-10037 | 1:500 |
| Chemical compound, drug | Goat anti- Rabbit IgG 568 | Invitrogen | A-11036 | 1:500 |
| Chemical compound, drug | Goat anti-Mouse IgG 488 | Invitrogen | A-11029 | 1:500 |
| Chemical compound, drug | Goat anti-Rabbit IgG 488 | Invitrogen | A-11034 | 1:500 |
| Chemical compound, drug | Hoechst | Sigma | 94403 | 1:1000 |
| Antibody | Anti-GFAP (Mouse Monoclonal) | Cell Signaling Technology | Cat. No. 3670S | 1:1000 |
| Antibody | Anti-β-III-tubulin (Rabbit Monoclonal) | Cell Signaling Technology | Cat. No. 5568S | 1:300 |
| Antibody | Anti-Nestin (Mouse Monoclonal) | Santa Cruz Biotechnology | Cat. No. Sc-23927 | 1:250 |
| Antibody | Anti-TSG101 (Rabbit Polyclonal) | ABclonal | Cat. No. A1692 | 1:1000 |
| Antibody | Anti-CD9 (Rabbit Monoclonal) | ABclonal | Cat. No. A19027 | 1:2000 |
| Antibody | Anti-CD63 (Rabbit Monoclonal) | ABclonal | Cat. No. A19023 | 1:1000 |
| Antibody | SYN1 (Rabbit Polyclonal) | Proteintech | Cat. No. 20258-1-AP | 1:1000 |
| Antibody | GAPDH (Mouse Monoclonal) | Proteintech | Cat. No. 60004-1-Ig | 1:1000 |

*Continued on next page*

*Continued*

| Reagent type (species) or resource | Designation | Source or reference | Identifiers | Additional information |
|---|---|---|---|---|
| Antibody | Caspase-3 (Mouse Monoclonal) | Proteintech | Cat. No. 66470-2-Ig | 1:1000 |
| Antibody | c-Caspase-3 (Rabbit Monoclonal) | Abcam | Cat. No. ab214430 | 1:500 |
| Antibody | STAT3 (Rabbit Monoclonal) | Abcam | Cat. No. ab68153 | 1:2000 |
| Antibody | PTPN1 (Rabbit Polyclonal) | Abclonal | Cat. No. A1590 | 1:1000 |
| Antibody | IKKα (Rabbit Monoclonal) | Abclonal | Cat. No. A19694 | 1:1000 |
| Antibody | β-Actin (Mouse Monoclonal) | Abclonal | Cat. No. AC004 | 1:2000 |
| Antibody | PSD95 (Rabbit Polyclonal) | Proteintech | Cat. No. 20665-1-AP | 1:200 |
| Chemical compound, drug | Goat anti-Mouse | Proteintech | Cat No. PR30012 | 1:5000 |
| Chemical compound, drug | Goat anti-Rabbit | Proteintech | Cat No. PR30011 | 1:5000 |
| Commercial assay or kit | ECL Enhanced Kit | Abclonal | RM00021 | |
| Chemical compound, drug | Isoflurane | RWD | R510-22-10 | |
| Chemical compound, drug | RIPA lysis buffer | Biosharp | BL504A | |
| Software, algorithm | Fiji | https://imagej.net/Fiji | N/A | |
| Commercial assay or kit | FD rapid Golgi Stain kit | FD Neuro Technologies | PK401A | |
| Commercial assay or kit | Nissl Stain Solution | Solarbio | G1434 | |
| Chemical compound, drug | DCF-DA | Invitrogen | D399 | 10 μM |
| Chemical compound, drug | 4% paraformaldehyde | Biosharp | BL539A | |
| Chemical compound, drug | Triton X-100 | Biofroxx | Cat. No. 1139 ML100 | |
| Chemical compound, drug | 10% formalin | Coolaber | SL1560 | |
| Chemical compound, drug | OCT compound | SAKURA | Japan | |
| Software, algorithm | Targetscan | http://www.targetscan.org/ | N/A | |
| Commercial assay or kit | Cell death detection kit | Roche | Cat. No. 11684795910 | |
| Commercial assay or kit | TBA method | Nanjing Jiancheng Bioengineering Institute | A003-1 | |
| Software, algorithm | miRcode | http://www.mircode.org/ | N/A | |
| Software, algorithm | miRDB | http://mirdb.org/ | N/A | |
| Software, algorithm | Cytoscape software | Cytoscape software | N/A | |
| Software, algorithm | RStudio | RStudio | N/A | |
| Software, algorithm | GraphPad PRISM | GraphPad Software | Version 9.2.0 | |

## Animals

Male C57BL/6 mice (age: 7–8 weeks, weight: 22–24 g) were selected due to estrogen and progesterone have recognized neuroprotective effects (*Newton et al., 2022*). All animal procedures were performed in compliance with guidelines for the care and use of animals and were approved by the University Huazhong Agriculture Institutional Animal Care and Use Committee (approval number: HZAUMO-2021-0111). Mice were assigned to MCAO/R or sham operation, and accepted NSCs or exosome treatment.

## NSCs induction and culture

NSCs induction was performed by iRegene Therapeutics, Wuhan, China as previously reported (*Cai et al., 2021*).Briefly, human iPSCs were cultured with STEMdiff Neural Induction Medium (iRegene Therapeutics). The medium was replaced daily until day 9. After the first passage, Y-27632 was added to the medium on day 1 to ensure the cell attachment and then removed from the medium on day 2. NSCs were cultured in STEMdiff Neural Progenitor Medium (iRegene Therapeutics) to maintain cell growth after passage. The identity of NSCs has been authenticated by immunofluorescence staining and qRT-PCR. Mycoplasma contamination is not detected in cell cultures.

## Exosomes isolation and detection

Cellular debris were removed from cell culture supernatant at $2000 \times g$ for 10 min. The supernatants were centrifuged at $20,000 \times g$ for 30 min. Then, exosomes were collected by ultracentrifugation (Beckman, America) at $100,000 \times g$ for 120 min. Finally, exosomes were washed in 12 ml PBS and collected again for 90 min. Exosomes were resuspended in PBS and protein concentration was measured by bicinchoninic acid (BCA) assay (Biosharp, China). For observation by transmission electron microscopy, exosomes were fixed in 2.5% glutaraldehyde at 4°C overnight and then mounted on a copper grid, stained with 2% uranyl acetate, and examined with a transmission electron microscope with 100 kV. For nanoparticle tracking analysis, exosomes were examined by Malvern Nano ZS90 as previously described (*Shi et al., 2018*). Exosomes were diluted in PBS and 1.0 ml suspension was loaded into a cuvette to measure and analyze.

## MCAO/R model

Ischemic stroke was established with MCAO/R surgery on male C57BL/6 mice (age: 7–8 weeks, weight: 22–24 g). Mice were anesthetized with 2% isoflurane (RWD, China). For focal cerebral ischemia, a silicon-coated filament (RWD, China) was inserted into the left middle cerebral artery to block blood flow. Sixty minutes later, the filament was extracted for reperfusion. Rectal temperature was maintained at 37°C during the entire procedure. Then anesthesia was discontinued and mice were allowed to recover. The cerebral blood flow (rCBF) was detected using a laser doppler flowmetry (Perimed, Sweden). A 55% decrease in the rCBF of the ipsilateral hemisphere, as compared to contralateral hemisphere, was considered the threshold for successful establishment of cerebral ischemia. Mice of the Sham group were performed the same as the MCAO/R procedure without filament insertion.

## Delivery of NSCs and exosomes

Eighty-eight MCAO/R mice were randomly divided into 4 groups at 7 days postoperation, and 12 mice with low body weight (less than 15 g) were excluded. Mice were anesthetized and placed in a mouse stereoscopic apparatus (RWD, China). The skull was drilled to make a burr hole above the lateral ventricle (AP+0, ML-1, DV-2.25 mm) for NSCs and exosomes injection. NSCs were genetically labeled with tdTomato for cell tracking. The five groups were treated as follows: PBS (MCAO/R mice treated with 5 µl PBS), Exo (MCAO/R mice treated with 10 µg exosomes in 5 µl PBS), NSC (MCAO/R mice treated with $5 \times 10^5$ NSCs in 5 µl PBS), NSC + Exo (MCAO/R mice treated with $5 \times 10^5$ NSCs combine with 10 µg exosomes in 5 µl PBS), and Sham (15 mice with sham operation not treated).

## Immunofluorescence staining

The cells were planted on round glass coverslips, and fixed with 10% formalin overnight at 4°C, then permeabilized with 0.25% Triton X-100, and blocked with 2% bovine serum albumin (BSA; Sigma, America) for 1 hr at room temperature. The coverslips were incubated with primary antibodies including anti-Sox2 (1:500, GeneTex, catalog 43019), anti-E-Cadherin (E-cad, 1:200, Proteintech, catalog 60335), anti-Pax6 (1:200, Proteintech, catalog 12323), anti-Ki67 (1:1000, Cell Signaling Technology, catalog 9449), anti-c-Caspase-3 (1:200, Abcam, catalog ab214430), and anti-MAP2 (1:500, Proteintech, catalog 67015) at 4°C overnight. The primary antibodies were then washed off and sections were incubated with secondary antibodies (Invitrogen, America) for 1 hr at room temperature. Cells were counterstained with Hoechst for 10 min after wash. Images were captured using a spinning disk confocal microscope (Andor Technology, UK).

Stroke leads to damage in the cerebral cortex, the atrophy was more severe without treatment, therefore we chose to observe the neuron recovery corresponding to the atrophied area in the model

group. For staining of mice brain tissues, mice were anesthetized and immediately perfused with PBS followed by 10% formalin for 30 min. Brains were fixed overnight in fixative at 4°C. Fixed brains were dehydrated in 30% sucrose in PBS for 2 days at 4°C. Brains were embedded in the optimal cutting temperature (OCT) compound (SAKURA, Japan). Brain sections were obtained at a thickness of 25 μm using a microtome cryostat (Leica, Germany). Tissues were permeabilized, blocked, and incubated as the above-mentioned protocol for cultured cell staining. Tissues were incubated with primary antibodies, including anti-NeuN (1:300, Abcam, catalog ab177487), anti-STEM121 (1:1000, Takara, catalog Y40410), anti-GFAP (1:1000, Cell Signaling Technology, catalog 3670S), anti-β-III-tubulin (1:300, Cell Signaling Technology, catalog 5568S), or anti-Nestin (1:250, Santa Cruz Biotechnology, catalog Sc-23927). For TUNEL staining, in situ cell death detection kit (Roche, Germany) was used to detect the cell apoptosis according to the manufacturer's instructions. Briefly, 3% BSA incubated sections were incubated with TUNEL reaction mixture for 1 hr at 37°C in the dark. Then sections were incubated with anti-NeuN primary antibody (1:300, Abcam, catalog ab177487) and corresponding secondary antibody successively.

## Western blot analysis

Total protein was extracted from NSCs or exosomes using RIPA lysis buffer (Biosharp, China) with protease inhibitor phenylmethylsulfonyl fluoride. Protein content was observed by the BCA assay (Biosharp, China). Protein samples (30 μg) were electrophoretically separated on 12% sodium dodecyl sulfate–polyacrylamide gel electrophoresis gels and then transferred to polyvinylidene fluoride membranes (Immobilon, America). The membranes were incubated with primary antibodies including TSG101, CD63, CD9, SYN1, GAPDH, β-actin, GFAP, STAT3, IKKα, and PTPN1 overnight at 4°C. The membranes were next incubated with secondary antibodies for 1 hr at room temperature (1:5000, Proteintech) and were detected using the ECL Enhanced Kit (ECL, Abclonal).

**Table 1.** Modified neurological severity score.

| Motor tests | Score |
| --- | --- |
| **Raise mouse by the tail** | 3 |
| Flexion of forelimb | 1 |
| Flexion of hindlimb | 1 |
| Head moved more than 10° to vertical axis | 1 |
| **Place mouse on the floor (minimum = 0; maximum = 3)** | 0–3 |
| Normal walk | 0 |
| Incapacity to walk straight | 1 |
| Circle toward the hemiplegia side | 2 |
| Fall down to the hemiplegia side | 3 |
| **Reflexes deficient and aberrant movements** | 1 |
| Seizures, myoclonus, or myodystony | 1 |
| **Beam balance tests (minimum = 0; maximum = 4)** | 4 |
| Maintain in stable posture | 0 |
| Hugs the beam and limb falls down from the beam | 1 |
| Hugs the beam or spins on beam (>40 s) | 2 |
| Attempts to balance on the beam but falls off (>20 s) | 3 |
| Fall off the beam (<20 s) | 4 |
| **Maximum points** | 10 |

## Motor function assessment

Testing on balance beam, ladder rung, rotarod test, and mNSS tasks was conducted preoperatively, and at 1–8 weeks postoperatively. Investigators were blinded to treatment groups in test.

### Balance beam

The balance beam apparatus used in this study was a 10-mm square wood in width and 50-cm wood in length (Beijing Zhongshi Science, China). Mice were trained to pass through the balance beam 3 days before the MCAO/R procedure. The mice that successfully passed the beam without foot slips were recruited and grouped. On behavioral test days (0, 4, and 8 weeks after treatment), the right feet slips were recorded when mice were passing through the balance beam three times. The scores were full score (10) minus the number of foot slips. If the mouse could not pass through or fall, the minimum score was recorded as 0.

### Ladder rung

The ladder rung instrument was made up of 2 transparent glass walls and 70 irregular metal bars (Beijing Cinontech Co Ltd, China). The number of mice that stepped wrong was recorded on behavioral test days. The score was calculated as 10 minus the number of wrong steps. If the mouse could not pass through, the minimum score was recorded as 0.

### Rotarod test

At 3 days before MCAO/R procedure, mice were trained on an accelerating rotarod at 30 rpm and only the mice that remained on the rotarod for 300 s at 30 rpm of three trials were recruited and grouped. The test was carried out at 30 rpm on behavioral days. The final scores were the seconds of mice remaining on the rotarod over three trials. The maximum score is 300 s.

### mNSS

According to the aforementioned report (*Table 1*; *Chen et al., 2001*), mNSS test is a composite of balance, motor, and reflex tests to assess neurological deficit. The normal score varies from 0 to 10, where 0 represents normal function and 10 maximal deficits. Three measurements were obtained per behavioral day.

## TTC staining

The brain of mice was removed quickly and carefully, then dissected into 2-mm thick sections on ice. Fresh brain slices were stained with 1% TTC solution (Solarbio, China) solution for 15 min at 37°C. TTC solution was then replaced with 4% paraformaldehyde and incubated overnight at 4°C. The sections were photographed with a digital camera.

## MRI

Mice were anesthetized and scanned by MRI (United Imaging, America) to detect the infarct area in the ipsilateral brain. Mice were imaged with a T2-weighted fast spin-echo imaging sequence using a 3T MRI scanner for mice.

## Golgi staining and analysis

Golgi staining was conducted using the FD rapid Golgi Stain kit (FD Neuro Technologies, America) according to the manufacturer's instructions. Mice were anesthetized and sacrificed, and the brains were removed quickly and immersed in the mixture of Solutions A and B for 2 weeks at room temperature in the dark. The brain was then transferred into Solution C for 48 hr at 4°C. Sections were cut with 100 µm thickness using a concussion slicer (Lecia, Germany) and stained with D and E mixture. Images were captured by an inverted microscope using Z-stack images (Lecia, Germany). Golgi-stained neurons were reconstructed using Fiji-Image J. The total dendritic length, the number of dendritic spines and intersections was calculated and analyzed by Sholl analysis according to the previous study (*Yang et al., 2020*).

**Table 2.** Quantitative PCR primer sequence for gene.

| Gene | Forward | Reverse |
|------|---------|---------|
| *mBax* | AGACAGGGGCCTTTTTGCTAC | AATTCGCCGGAGACACTCG |
| *mGapdh* | AGGTCGGTGTGAACGGATTTG | GGGGTCGTTGATGGCAACA |
| *mTnfa* | CAGGCGGTGCCTATGTCTC | CGATCACCCCGAAGTTCAGTAG |
| *mIl1b* | GCCCATCCTCTGTGACTCAT | AGCTCATATGGGTCCGACAG |
| *mIl10* | CTTACTGACTGGCATGAGGATCA | GCAGCTCTAGGAGCATGTGG |
| *mC3* | GAGCGAAGAGACCATCGTACT | TCTTTAGGAAGTCTTGCACAGTG |
| *mGbp2* | CTGCACTATGTGACGGAGCTA | CGGAATCGTCTACCCCACTC |
| *mLcn2* | GCAGGTGGTACGTTGTGGG | CTCTTGTAGCTCATAGATGGTGC |
| *mStat3* | CACCTTGGATTGAGAGTCAAGAC | AGGAATCGGCTATATTGCTGGT |
| *mPtpn1* | GTCGGATTAAATTGCACCAGGA | TGATGCGGTTGAGCATGACC |
| *mChuk* | GGTTTCGGGAACGTCAGTCTG | GCACCATCGCTCTCTGTTTTT |
| *hNQO1* | GAAGAGCACTGATCGTACTGGC | GGATACTGAAAGTTCGCAGGG |
| *hNRF2* | CCTGTAAGTCCTGGTCATCG | TTTCTACAGGGAATGGGATA |
| *hCHOP* | GGAAACAGAGTGGTCATTCCC | CTGCTTGAGCCGTTCATTCTC |
| *hSOD2* | GCTCCGGTTTTGGGGTATCTG | GCGTTGATGTGAGGTTCCAG |
| *hSTAT3* | CAGCAGCTTGACACACGGTA | AAACACCAAAGTGGCATGTGA |
| *hPTPN1* | GCAGATCGACAAGTCCGGG | GCCACTCTACATGGGAAGTCAC |
| *hCHUK* | GGCTTCGGGAACGTCTGTC | TTTGGTACTTAGCTCTAGGCGA |
| *hGAPDH* | GGAGCGAGATCCCTCCAAAAT | GGCTGTTGTCATACTTCTCATGG |

## Nissl staining

Nissl staining was conducted using the Nissl Stain Solution (Solarbio, America) according to the manufacturer's instructions. Mice brain sections were stained with methylene blue stain for 10 min at 65°C, then differentiated by nissl differentiation solution for 3 min. The brain sections were subsequently treated in ammonium molybdate solution for 5 min followed by a quick rinse quickly in distilled water to avoid decolorizing. Images were taken using an inverted microscope (Lecia, Germany).

## qRT-PCR

Total RNA was extracted from ipsilateral brain tissue using TriQuick Reagent (Solarbio, China). Reverse transcription was performed by HiFiScript gDNA Removal RT MasterMix kit (CWBIO, China) following the manufacturer's instructions. qRT-PCR was implemented in CFX (Bio-rad) using MagicSYBR Mixture (CWBIO, China). The threshold cycle (CT) was evaluated to quantify transcripts. The relative expression level of a specific gene was calculated as the expression $2^{-\Delta\Delta CT}$. The primers used in this study are listed in *Table 2*. To examine the expression of miRNAs, total miRNA was extracted from cells by miRcute miRNA Isolation Kit (TIANGEN, China), and reverse transcription was performed by miRcute Plus miRNA First-Strand cDNA Kit (TIANGEN, China). qRT-PCR was implemented by miRcute Plus miRNA qPCR Kit (SYBR Green, TIANGEN, China). The CT value was used to quantify transcripts. The relative expression level of a specific miRNA was calculated as the expression $2^{-\Delta\Delta CT}$. The forward primers used in this study are shown in *Table 3*.

## Isolation and culture of primary astrocytes

Primary astrocytes were prepared from cerebral cortices of 1-day-old neonatal C57BL/6 mice. Cerebral cortices were isolated carefully and digested with 0.125% trypsin at 37°C for 10 min followed by filtering through a 70-μm cell strainer. The isolated cells were cultured in Dulbecco's modified of Eagle's medium/F12 medium for 10 days until they reached 80% confluence.

**Table 3.** Quantitative PCR primer sequence for miRNA.

| Gene | Forward |
|---|---|
| hsa-miR-9-5p | UCUUUGGUUAUCUAGCUGUAUGA |
| hsa-miR-7-5p | UGGAAGACUAGUGAUUUUGUUGUU |
| hsa-miR-148a-3p | UCAGUGCACUACAGAACUUUGU |
| hsa-miR-26a-5p | UUCAAGUAAUCCAGGAUAGGCU |
| hsa-miR-21-5p | UAGCUUAUCAGACUGAUGUUGA |
| hsa-miR-151a-3p | CUAGACUGAAGCUCCUUGAGG |
| hsa-miR-218-5p | UUGUGCUUGAUCUAACCAUGU |
| hsa-miR-30a-5p | UGUAAACAUCCUCGACUGGAAG |
| hsa-let-7i-5p | UGAGGUAGUAGUUUGUGCUGUU |
| hsa-miR-25-3p | CAUUGCACUUGUCUCGGUCUGA |

## Oxygen-glucose deprivation and reoxygenation

To perform OGD/R on cultured NSCs, HT22 cells and primary astrocytes, the normal culture medium was replaced with Dulbecco's modified of Eagle's medium (Solarbio, China). The culture was then incubated in a hypoxia chamber aerated with 5% $CO_2$, 94% $N_2$, and 1% $O_2$ at 37°C for 2 hr for NSCs and HT22 cells, and 6 hr for primary astrocytes. Then the NSCs were transferred back into the normal culture medium and incubated in normal culture conditions for 24 hr.

## Intracellular ROS detection

ROS level was detected using the fluorescent probe DCF-DA (Invitrogen, America). Cultured NSCs were incubated with 10 µM DCF-DA for 30 min at 37°C and then fixed with 4% paraformaldehyde. DCF-DA fluorescence was photographed and quantified via a spinning disk confocal microscope (Andor Technology, UK).

## MDA-level measurement

The MDA level was measured at 3 days after treatment by the TBA method (Nanjing Jiancheng Bioengineering Institute, China) according to the manufacturer's instructions. The ipsilateral brain was homogenized and incubated in the assay solution at 95°C for 80 min. The optical density was measured at 532 nm by a microplate reader. The value was calculated based on the standard formula.

## Microarray analysis of exosomal miRNAs

Sequencing libraries of miRNAs of NSC-derived exosomes were produced using NEBNext Multiplex Small RNA Library Prep Set for Illumina (NEB, United States) following the previous report (**Cai et al., 2021**).

The downstream target genes of exosomal miRNAs were predicted using three online databases: Targetscan (http://www.targetscan.org/), miRcode (http://www.mircode.org/), and miRDB (http://mirdb.org/). The enrichment analysis of the predicted target genes was performed using Cluster-Profiler R package for GO process and KEGG pathway enrichment. The miRNA–mRNA regulatory network was built by Cytoscape software. miRNA was calculated by qRT-PCR. Libraries were prepared by ligating adaptors to the total RNA, PCR amplification and size selection using 6% polyacrylamide gels. Sequencing was performed on Illumina NovaSeq 6000 (Illumina Inc, USA).

## Statistics

GraphPad Prism version 7 was used for statistical analyses. Unpaired *t*-tests (two-tailed) were used for single comparisons, and two-way analysis of variance was used for multiple comparisons. Survival analysis was performed via the Kaplan–Maier method. All data are presented as mean ± standard error of the mean.

## Acknowledgements

This work is supported by Key Project of Research and Development of Hubei Province （Grant No. 2022BCE049）, National Natural Science Foundation of China (Grant No. 32070973, 31871481), Fundamental Research Funds for the Central Universities (Program No. 2662022JC002).

## Additional information

### Competing interests

Weibing Mao, Lumeng Niu, Yasha Zhu, Meng Cai, Jun Wei: is affiliated with iRegene Therapeutics Co., Ltd. The author has no financial interests to declare. The other authors declare that no competing interests exist.

### Funding

| Funder | Grant reference number | Author |
| --- | --- | --- |
| Key Project of Research and Development of Hubei Province | 2022BCE049 | Zhiqiang Dong |
| National Natural Science Foundation of China | 32070973 | Zhiqiang Dong |
| National Natural Science Foundation of China | 31871481 | Zhiqiang Dong |
| Fundamental Research Funds for the Central Universities | 2662022JC002 | Zhiqiang Dong |

The funders had no role in study design, data collection, and interpretation, or the decision to submit the work for publication.

### Author contributions

Ruolin Zhang, Conceptualization, Resources, Data curation, Software, Formal analysis, Validation, Investigation, Visualization, Methodology, Writing - original draft, Writing – review and editing; Weibing Mao, Lumeng Niu, Validation, Methodology; Wendai Bao, Conceptualization, Writing – review and editing; Yiqi Wang, Software, Visualization; Ying Wang, Zhihao Yang, Haikun Song, Investigation, Methodology; Yasha Zhu, Guangqiang Li, Methodology; Jincao Chen, Min Zhang, Conceptualization; Jiawen Dong, Zilong Yuan, Validation; Meng Cai, Nanxiang Xiong, Project administration, Writing – review and editing; Jun Wei, Conceptualization, Supervision, Project administration, Writing – review and editing; Zhiqiang Dong, Conceptualization, Supervision, Funding acquisition, Writing - original draft, Project administration, Writing – review and editing

### Author ORCIDs

Ruolin Zhang  http://orcid.org/0000-0003-0462-0955
Jun Wei  http://orcid.org/0009-0004-7233-3784
Zhiqiang Dong  http://orcid.org/0000-0001-6259-915X

### Ethics

All animal procedures were performed in compliance with guidelines for the care and use of animals and were approved by the University Huazhong Agriculture Institutional Animal Care and Use Committee (approval number: HZAUMO-2021-0111). Mice were assigned to MCAO/R or sham operation, and accepted NSCs or exosome treatment. All surgery was performed under sodium pentobarbital anesthesia, and every effort was made to minimize suffering.

### Decision letter and Author response

Decision letter https://doi.org/10.7554/eLife.84493.sa1
Author response https://doi.org/10.7554/eLife.84493.sa2

## Additional files

### Supplementary files
• MDAR checklist

### Data availability
All data generated or analyzed during this study are included in the manuscript and supporting file. Figure 1—Source Data 1, Figure 2—Source Data 1,Figure 3—Source Data 1 and Figure 4—Source Data 1, Figure 1—supplement 1—Source Data 1 and 2, Figure 2—supplement 1—Source Data 1, Figure 4—supplement —Source Data 1 and Figure 5—supplement 1—Source Data 1 provided for Figures 1 to 5 and supplement figure 1-5.Code for figure 5 is uploaded as Source Code Files 1-4.The sequencing data of exosomes have been deposited in GEO under accession codes GSE217074.

The following dataset was generated:

| Author(s) | Year | Dataset title | Dataset URL | Database and Identifier |
|---|---|---|---|---|
| Cai M | 2023 | NSC-derived exosomes enhance therapeutic effects of NSC transplantation on cerebral ischemia in mice | https://www.ncbi.nlm.nih.gov/geo/query/acc.cgi?acc=GSE217074 | NCBI Gene Expression Omnibus, GSE217074 |

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
