## [Editor Report]

The authors have demonstrated that NSC-derived exosomes could act as a supportive adjuvant for NSC transplantation after stroke. NSC-derived exosomes significantly reduced the inflammatory response, alleviated oxidative stress after NSC transplantation, and facilitated NSC differentiation in vivo. Overall, the combination of NSCs with exosomes ameliorated the injury of brain tissue including cerebral infarction, neuronal death, and glial scarring, and promoted the motor function recovery.

---

## [Decision Letter]

**Decision letter after peer review:**

Thank you for submitting your article "NSC-derived exosomes enhance therapeutic effects of NSC transplantation on cerebral ischemia in mice" for consideration by *eLife*. Your article has been reviewed by 3 peer reviewers, including Keqiang Ye as Reviewing Editor and Reviewer #1, and the evaluation has been overseen by Lu Chen as the Senior Editor. The following individuals involved in review of your submission have agreed to reveal their identity: Zhengang Yang (Reviewer #2); Dandan Sun (Reviewer #3).

This study investigates efficacy of exosomes from human (iPSCs) in improving NSC therapy for neuroprotection in mouse stroke model. One-week post-stroke, administration of NSCs through lateral ventricle injections in combination with exosomes significantly improved post-stroke survival, neurological function recovery and brain lesion attenuation in mice at 8-week post treatment. The strengths of this study include the positive outcomes from this combinatory treatment delivered at the subacute phase and multiple assessments of neurological function impairments; Notably, it is non-invasive, unbiased assessment of brain lesion with MRI. However, the evaluation of the possible underlying mechanisms is weak, which is articulated in each individual reviewer's comment. Extensive revision is necessary for strengthening this manuscript.

Essential revisions:

1) Biochemical characterization of the observed IF and IHC events.

2) Additional evidence about the cell death of the transplanted cells. The immunostaining of the Caspase-3 or TUNEL staining should be used to address this issue.

3) The evaluation of the possible underlying mechanisms is weak, which included reduction of reactive astrocytes, increased NeuN^+^ cells, and possible roles of anti-inflammatory miRNA profiles of exosomes from NSCs in the study.

*Reviewer #1 (Recommendations for the authors):*

Employing ischemic stroke in mice with MCAO/R, the authors tested different treatment strategies using transplantation of iPSC-induced NSCs and NSC-derived exosomes. They showed that NSC-derived exosomes could promote NSCs differentiation, reduce the oxidative stress and inflammation, and alleviate the formation of glial scars after ischemia and reperfusion, and as a result, could enhance the therapeutic effects of NSC transplantation. The authors further explored the molecular mechanisms through profiling the miRNAs of the NSC-derived exosomes. However, the signalings impacted by the miRNAs were not biochemically validated by immunoblotting or immunofluorescent staining. This is crucial for confirming the actual molecules are indeed influenced by the miRNA as presented in Figure 5. With this evidence, the main conclusion will be highly strengthened.

*Reviewer #2 (Recommendations for the authors):*

Methods need to be further clarified.

*Reviewer #3 (Recommendations for the authors):*

There are some concerns and suggestions regarding the study.

1. The model of 60-min MCAO in mice, surprisingly, produced >50% mortality at 2-week while the infarct volume was significantly reduced at 7- day post stroke (Supplemental Figure 1). In addition, Rotarod and mNSS tests of PBS-stroke mice showed, relatively, no recovery by 8-week post-stroke. These data are different from the reported general outcomes in mice using this model (<25% mortality, recovery of sensorimotor functions at 4-week post-stroke, etc). In addition, supplemental Figure 4 shows that the infarct core was not centered at the middle cerebral artery territory. The authors need to evaluate these issues and determine that appropriate surgical procedures of MCAO have been accomplished.

2. Since most outcomes were assessed at 8-week post stroke, it is important to address whether exosomes of neuronal stem cells (NSC) derived from human iPSCs improves NSC therapy via reduction of neuronal death, increases of neurogenesis? or through neuron-independent mechanisms. The authors failed to conduct these experiments.

3. Figure 4 of cell culture study and Figure 5 of exosome miRNA profiling provided no direct support to the in vivo study nor the final conclusions.

4. changes of migration of NSCs with or without exosomes should be assessed.

---

## [Author Response]

Essential revisions:1) Biochemical characterization of the observed IF and IHC events.

We have performed biochemical characterization of the observed IF and IHC events and added the following data to the results: the expression of neuronal markers, including RBFOX3/NeuN by qRT-PCR (Figure 2 – supplement 1A), SYN1 by western blot (Figure 2H); the expression of GFAP by Western blot (Figure 4G). We have modified the text accordingly to describe these data (Page 6, Line 121-124; Page 7, Line 146-147; Page 8, Line 148-151 and Page 11, Line 227-229).

2) Additional evidence about the cell death of the transplanted cells. The immunostaining of the Caspase-3 or TUNEL staining should be used to address this issue.

We thank the reviewers for the constructive comments. We have conducted immunostaining of Caspase-3 at 7 days after transplantation using the human-specific STEM121 antibody to demonstrate the transplanted cells. We have added the results to Figure 3A and modified the text accordingly (Page 8, Line 156-165).

3) The evaluation of the possible underlying mechanisms is weak, which included reduction of reactive astrocytes, increased NeoN+ cells, and possible roles of anti-inflammatory miRNA profiles of exosomes from NSCs in the study.

We have performed more experiments to explore and gained more insights into the underlying mechanisms.

A1 reactive astrocytes have been reported form glial scars to inhibit the neuronal recovery at the chronic stage after stroke (Clarke et al., 2018). Our data suggested that the reactive astrocytes decreased at the chronic stage after exosomes treatment. We hypothysize that exosomes inhibit the activation of A1 reactive astrocytes. We have examined the activation of A1 subtype in astrocytes following OGD/R and exosomes treatment at 24 h and 48 h (Figure 4 – supplement 1B and 1C), which confirmed our hypothesis. Therefore, we further determined the subtype of reactive astrocytes that underlying the effects of NSC+Exo treatment.

We inspected the effect of exosomes on the survival of neurons by examining the expression of apoptotic marker cleaved Caspase-3 (c-Caspase-3) and pro-apoptotic gene Bax in HT22 cells after OGD/R and exosome treatment. The results showed that exosomes reduced the OGD/R induced neuronal apoptosis (Figure 2 – supplement 1C-E, Page 7, Line 128-137). Besides the neuronal apoptosis, we evaluated the recovery of neural connection by checking the expression of SYN1 in mice of different groups and confirmed that SYN1 expression was increased in NSC+Exo group compared with NSC group (Figure 2H). These new data suggest that NSC-drived exosomes might promote the therapeutic effects of NSCs transplantation through inhibiting the apoptosis of neurons and assisting the recovery of neuronal connection.

As mentioned above, we examined the effects of exosomes on the survival of transplanted cells by immunostaining of Caspase-3 and STEM121 at 7 days after transplantation (Figure 3A, Page 8, Line 156-165). Furthermore, our additional data revealed that exosomes increased the differentiation of transplanted NSCs using immunostaining of RBFOX3/NeuN, PSD95 (Figure 3B; Figure 3 – supplement 1D, Page 9, Line 175-183). Interestingly, the anti-STEM121 staining revealed a larger distribution area of STEM121^+^ cells in NSC+Exo group compared with NSC group (Figure 3 – supplement 1C, Page 8, Line 165-168) suggesting that exosomes might help the migration of transplanted cells.

Reviewer #1 (Recommendations for the authors):Employing ischemic stroke in mice with MCAO/R, the authors tested different treatment strategies using transplantation of iPSC-induced NSCs and NSC-derived exosomes. They showed that NSC-derived exosomes could promote NSCs differentiation, reduce the oxidative stress and inflammation, and alleviate the formation of glial scars after ischemia and reperfusion, and as a result, could enhance the therapeutic effects of NSC transplantation. The authors further explored the molecular mechanisms through profiling the miRNAs of the NSC-derived exosomes. However, the signalings impacted by the miRNAs were not biochemically validated by immunoblotting or immunofluorescent staining. This is crucial for confirming the actual molecules are indeed influenced by the miRNA as presented in Figure 5. With this evidence, the main conclusion will be highly strengthened.

We thank the reviewers for the constructive comments. We have biochemically validated the effects of exosome treatment on the protein expression of target genes *STAT3*, *PTPN1* and *CHUK* using western blot in MCAO/R mice. The results confirmed that exosome treatment reduced the expression of downstream genes in brain tissues (Figure 5 – supplement 1C, Page 12, Line 247-251; Page 13, Line 252-254), suggesting that exosomes might modulate the recipient cells and brain tissues through carrying miRNAs that regulate the expression of target genes.

Reviewer #2 (Recommendations for the authors):Methods need to be further clarified.

We have modified the Methods part according to the reviewer’s suggestion.

Reviewer #3 (Recommendations for the authors):There are some concerns and suggestions regarding the study.1. The model of 60-min MCAO in mice, surprisingly, produced >50% mortality at 2-week while the infarct volume was significantly reduced at 7- day post stroke (Supplemental Figure 1). In addition, Rotarod and mNSS tests of PBS-stroke mice showed, relatively, no recovery by 8-week post-stroke. These data are different from the reported general outcomes in mice using this model (<25% mortality, recovery of sensorimotor functions at 4-week post-stroke, etc). In addition, supplemental Figure 4 shows that the infarct core was not centered at the middle cerebral artery territory. The authors need to evaluate these issues and determine that appropriate surgical procedures of MCAO have been accomplished.

We thoroughly searched the literature of the MCAO/R methods and chose the thread occlusion to establish cerebral ischemia. The degree of ischemia and injury during surgery may vary, which could be the reason for different mortality rates. For example, it is reported that the mortality rate could be even higher (80%-100%) with the same thread occlusion MCAO/R method (Yuan *et al.,* 2018, Jiang *et al.,* 2019).

In order to test the therapeutic effects of NSCs and exsomome treatment, we adopted settings with higher difficulty for the behavioral experiments, for example, 30 rpm for the rotarod test. Previous study also confirmed much delayed behavioral recovery with similar settings, in which the MCAO mice did not show significant recovery in the rotarod test for 7 weeks after surgery (Wang *et al.,* 2020).

Our results showed that the infarct core caused by thread occlusion MCAO/R is in the middle-lower region on the coronal section of cerebral cortex, which is actually consistent with previous reports (Bouët et al., 2007).

2. Since most outcomes were assessed at 8-week post stroke, it is important to address whether exosomes of neuronal stem cells (NSC) derived from human iPSCs improves NSC therapy via reduction of neuronal death, increases of neurogenesis? or through neuron-independent mechanisms. The authors failed to conduct these experiments.

We thank the reviewer for the constructive comments. We have performed more experiments and verified that exosomes could reduce the death of transplanted cells (please also see response to reviewer #2-major points #1) and increase the generation of mature neurons and synapses by the transplanted cells (please also see response to reviewer #2-major points #2). However, there may be neuron-independent mechanisms for the effects of exosomes, which demands further exploration.

3. Figure 4 of cell culture study and Figure 5 of exosome miRNA profiling provided no direct support to the in vivo study nor the final conclusions.

Oxidative stress is one of the major challenges for the survival and colonization of transplanted NSCs. Our cell culture study disclosed that exosome treatment could reduce the oxidative stress of NSCs, which is consistent of our in vivo study that MDA content was significantly decreased in exosome-treated mice. With both in vivo and in vitro evidence, we have a solid foundation to hypothesize that exosome could enhance the therapeutic effects of NScs by reducing oxidative stress.

Besides the in vitro data on the candidate target genes of exosomal miRNA, We have examined the effects of exosome treatment on the protein expression of target genes *STAT3*, *PTPN1* and *CHUK* using western blot in MCAO/R mice. The results confirmed that exosome treatment reduced the expression of downstream genes in brain tissues (Figure 5 – supplement 1C, Page 12, Line 247-251; Page 13, Line 252-254), which provides clues for the mechanisms of NSCs+Exo effects, that is, exosomes modulating the recipient cells and brain tissues through carrying miRNAs that regulate the expression of target genes.

4. changes of migration of NSCs with or without exosomes should be assessed.

We have examined the migration of NSCs with or without exosomes at 7 days post transplantation, and the results suggest that exosomes might help the migration of transplanted cells (Figure 3 – supplement 1C, Page 8, Line 165-168).

References

Bouët, V., T. Freret, J. Toutain, D. Divoux, M. Boulouard, and P. Schumann-Bard. 2007. Sensorimotor and cognitive deficits after transient middle cerebral artery occlusion in the mouse. Exp Neurol 203:555-567. DOI:https://doi.org/10.1016/j.expneurol.2006.09.006, PMID: 17067578

Clarke, L. E., S. A. Liddelow, C. Chakraborty, A. E. Münch, M. Heiman, and B. A. Barres. 2018. Normal aging induces A1-like astrocyte reactivity. Proc Natl Acad Sci U S A 115:E1896-e1905. DOI:https://doi.org/10.1073/pnas.1800165115, PMID: 29437957

Jiang, X. C., J. J. Xiang, H. H. Wu, T. Y. Zhang, D. P. Zhang, Q. H. Xu, X. L. Huang, X. L. Kong, J. H. Sun, Y. L. Hu, K. Li, Y. Tabata, Y. Q. Shen, and J. Q. Gao. 2019. Neural Stem Cells Transfected with Reactive Oxygen Species-Responsive Polyplexes for Effective Treatment of Ischemic Stroke. Adv Mater 31:e1807591. DOI:https://doi.org/10.1002/adma.201807591, PMID: 30633395

Wang, R., H. Pu, Q. Ye, M. Jiang, J. Chen, J. Zhao, S. Li, Y. Liu, X. Hu, M. Rocha, A. P. Jadhav, J. Chen, and Y. Shi. 2020. Transforming Growth Factor Β-Activated Kinase 1-Dependent Microglial and Macrophage Responses Aggravate Long-Term Outcomes After Ischemic Stroke. Stroke 51:975-985. DOI:https://doi.org/10.1161/strokeaha.119.028398, PMID: 32078472

Yuan, D., C. Liu, J. Wu, and B. Hu. 2018. Nest-building activity as a reproducible and long-term stroke deficit test in a mouse model of stroke. Brain Behav 8:e00993. DOI:https://doi.org/10.1002/brb3.993, PMID: 30106254